# Glutamine for Amelioration of Radiation and Chemotherapy Associated Mucositis during Cancer Therapy

**DOI:** 10.3390/nu12061675

**Published:** 2020-06-04

**Authors:** Peter M. Anderson, Rajesh V. Lalla

**Affiliations:** 1Cleveland Clinic Pediatric Hematology/Oncology and Bone Marrow Transplant, Pediatric Institute and Taussig Cancer Institute, Cleveland, OH 44195, USA; 2UConn Health, School of Dental Medicine, Farmington, CT 06030 USA; lalla@uchc.edu

**Keywords:** glutamine, trehalose, mucositis, radiation injury, chemotherapy associated malnutrition, glutathione, amino acid supplementation in cancer

## Abstract

Glutamine is a major dietary amino acid that is both a fuel and nitrogen donor for healing tissues damaged by chemotherapy and radiation. Evidence supports the benefit of oral (enteral) glutamine to reduce symptoms and improve and/or maintain quality of life of cancer patients. Benefits include not only better nutrition, but also decreased mucosal damage (mucositis, stomatitis, pharyngitis, esophagitis, and enteritis). Glutamine supplementation in a high protein diet (10 grams/day) + disaccharides, such as sucrose and/or trehalose, is a combination that increases glutamine uptake by mucosal cells. This increased topical effect can reduce painful mucosal symptoms and ulceration associated with chemotherapy and radiation in the head and neck region, esophagus, stomach and small intestine. Topical and oral glutamine seem to be the preferred routes for this amino acid to promote mucosal healing during and after cancer treatment.

## 1. What Is Glutamine

Glutamine is an L-alpha-amino acid. It is the most abundant free amino acid in human blood. Glutamine is needed for several functions in the body including for the synthesis of proteins as well as an energy source. Glutamine can be synthesized by the body and can also be obtained from the diet if needed. 

## 2. Importance of Glutamine 

Glutamine is an important nitrogen donor in intracellular metabolism and in the maintenance of intestinal tract, immune cells, and muscle [1,2,3,4,5,6]. Weight loss in cancer patients is common, but sarcopenia (loss of muscle mass) is associated with increased complications and significantly worse survival [7,8,9,10]. Glutamine is a preferred fuel for both lymphocytes [11,12] and gastrointestinal (GI) tract [1], thus it plays an important role in helping to defend against infections and to assist mucosa in being a barrier against infection (Figure 1). Glutamine has a central role in intracellular metabolism and acts as a nitrogen shuttle between muscle and other tissues; it is at a high and relatively stable concentration in plasma and red blood cells and at a much higher concentration in muscle compared to other amino acids [1,13,14,15,16]. Since plasma glutamine concentrations are only minimally affected over time by either glutamine ingestion or infusion, muscle can be considered as a “bank” and the liver can be considered as the “banker” (Figure 2) [14]. 

## 3. Oral Mucositis from Cancer Therapy: A Common Problem

Oral Mucositis (OM) refers to inflammation and ulceration of the oral mucosa as a side-effect of cancer therapy [17,18,19,20,21,22,23,24,25]. OM and esophagitis can occur secondary to systemic chemotherapy for cancer [26,27,28], high-dose chemotherapy as a hematopoietic transplant preparative regimen [27,29,30,31,32,33,34,35,36,37,38,39] or due to radiation therapy (RT) for head and neck (H&N) cancer [29,40,41,42,43,44] or if the oropharynx or esophagus is in field during and after radiation of bone metastases. OM is a common problem and occurs in about 20–40% of patients receiving conventional chemotherapy for solid tumors, about 80% of patients receiving high dose chemotherapy prior to a hematopoietic stem cell transplantation (HSCT) and almost all patients receiving RT for H&N cancer [19,20,21]. Ulcerative OM and esophagitis are extremely painful, with many patients needing systemic opioids such as morphine or fentanyl for pain management [21,27,45,46,47]. 

The intense mouth and esophageal pain from mucosal injury as well as enteritis results in significant nutritional compromise, which can lead to weight loss, impaired healing, and decreased resistance to infection [2,48,49]. Because of malnutrition, feeding through a gastrostomy tube or total parenteral nutrition may be needed [21,45]. Quality of life is markedly reduced [50,51]. Maintenance of oral hygiene becomes difficult. The oral ulcerations can get secondarily infected, which may lead to bacteremia in patients with immunosuppression due to high dose chemotherapy. Cost of care is increased due to costs associated with pain management, nutritional support, infection control, and additional hospitalization. Perhaps most importantly, severe OM can necessitate a reduction in chemotherapy dosage or a break in RT and may discourage patients from getting therapy in a timely manner or at all, potentially affecting the overall success of cancer therapy [52,53,54,55,56,57,58,59,60]. 

The pathogenesis of OM is complex and multifactorial (Figure 1) [18,19,21,45,52,53,54,55,56,57,58,59]. The primary etiology is direct damage from chemotherapy or RT to the oral epithelium. Table 1 lists common chemotherapy agents and how these often cause cytopenias and GI side effects including nausea, malnutrition, and mucosal damage (mucositis, stomatitis, esophagitis, enteritis) independently. In particular, the basal cells of the oral epithelium become unable to replenish the mucosa by their normally rapid division [50]. The cancer therapy also causes a variety of additional effects in the oral, esophageal, and intestinal mucosa, including activation of various inflammatory pathways, leading to up-regulation of inflammatory cytokines and other tissue-damaging molecules [57,58,59,60,61,62]. Once ulceration develops, the lesions become colonized by the local flora which can further aggravate the injury and impair healing. The overall result is a sustained period of oral and/or alimentary canal damage, inflammation and sometimes ulceration, with healing occurring one to several weeks after radiation and/or chemotherapy. If healing is slow, if may be difficult to sustain the intended schedule of chemotherapy or radiation therapy as illustrated in Figure 1. Conversely, accelerated healing can not only improve quality of life (QOL) but also facilitate timely cancer treatment with fewer secondary therapy-related side effects. Development of a “therapeutic alliance” to get feedback and provide options, to make each cycle of chemotherapy better and avoid a catabolic state, is an important role of, not only the oncologist, but also caregivers, dieticians, and nurses [60]. 

## 4. Suggested Use of Glutamine in Mucositis Management from MASCC/ISOO 2019 Guidelines

A number of clinical studies have evaluated the use of glutamine to prevent or treat OM in various cancer populations. The Multinational Association of Supportive Care in Cancer and International Society of Oral Oncology (MASCC/ISOO) recently published systematic reviews of the literature on various interventions for OM and evidence-based clinical practice guidelines [18,20,45,63,64,65]. These include newer guidelines related to glutamine [65]. In patients receiving concurrent chemotherapy and radiation for H&N cancer, a suggestion was made in favor of the use of oral (PO) and/or swish and spit glutamine for the prevention of OM. This suggestion was based on Level II evidence, which was derived from two randomized controlled trials (RCTs). In these studies, the use of oral glutamine resulted in a significant reduction of severity of OM [66,67]. In addition, it also reduced the duration of OM in one study and the associated pain in another study. In one of these studies of H&N cancer patients, the glutamine was delivered as a “swish and swallow” liquid formulation, which indicates the possibility of a topical effect as well [27,66,68].

There are also multiple studies supporting use of oral (PO) glutamine or parenteral for OM in other populations including patients with solid cancers and patients receiving HSCT [27,28,68,69]. These studies are summarized by Yarom et al. [65]. However, due to inadequate or conflicting data in these groups, no MASCC/ISOO guideline was possible for oral glutamine in these other populations.

On the other hand, in patients undergoing HSCT, the guidelines include a recommendation against the use of parenteral (intravenous) glutamine for the prevention of OM. This recommendation was based on the results of six RCTs, of which two found parenteral glutamine to be effective and four did not demonstrate a beneficial effect [65]. Furthermore, in one study of HSCT patients, a correlation of parenteral glutamine treatment with relapse and mortality was documented [69]. Taking into account all the above factors, the panel recommended against the use of parenteral glutamine for prevention of OM in patients undergoing HSCT. 

The above results suggest that oral glutamine can be beneficial for management of OM and that there could be a positive topical effect. This approach is supported by results of a pilot RCT in which glutamine used topically as a mouth rinse and not swallowed was found to be effective in reducing severity and duration of OM [42]. 

## 5. Potential Mechanisms of Glutamine Activity in Mucositis Management 

Neutrophils, macrophages, and lymphocytes are needed for mucosal barrier immune defenses. Since glutamine is fuel for leukocytes, topical/oral/enteral glutamine may contribute to mucosal healing by not only a direct effect on mucosal epithelial cells, but also by improvement in host mucosal immune function and ability to resist microbial invasion [2,3,5,6,11,12,70].

Interestingly, resilience of lymphocyte recovery, as measured by absolute lymphocyte count (ALC) after the very first cycle of chemotherapy, has been associated with a better prognosis in a variety of malignancies including acute lymphoblastic leukemia as well as tumors such as osteosarcoma, Ewing sarcoma, and rhabdomyosarcoma [71,72,73,74,75,76,77]. It is possible that better nutrition, with amino acids including glutamine as fuel for lymphocytes, could contribute to ALC recovery and/or resilience. Animal models and human studies have shown glutamine supplementation improves the ability to resist the toxic effects of radiation to the GI tract [78,79,80,81,82,83]. 

Detoxification and resilience to free radical damage by chemotherapy (e.g., doxorubicin or cyclophosphamide) and/or radiation of normal tissues and tumors can involve the antioxidant glutathione. Since glutamine is a substrate for glutathione synthesis, adequate mucosal cell glutamine may contribute towards improved healing after chemotherapy and radiation damage [84,85] as well as, interestingly, the simultaneous inhibition of glutathione levels in tumors, too [86,87,88,89]. Furthermore, decreased inflammatory cytokines in normal cells and increased pro-apoptosis proteins in cancer cells were observed with glutamine + disaccharide supplementation [89]. Thus, glutamine can contribute to selective improvement in host cell resilience, less inflammation, and decreased ability of tumors to detoxify chemotherapy or resist radiation, i.e. an improved therapeutic index of the anti-cancer therapy. 

Finally, fewer complications and improved survival have been seen in cancer patients that have less muscle loss, also known as sarcopenia [7,8,10]. This underscores the importance of enteral nutrition and activity to maintain muscle mass rather than having muscle become a back-up system to provide amino acids including glutamine in catabolic oncology patients. Figure 2 illustrates this dynamic physiology and the importance of glutamine in enteral nutrition in cancer patients.

## 6. Bioavailability of Oral Glutamine Locally vs. Systemically in Mucositis Management 

Although glutamine is easily the most abundant plasma amino acid (~20%; [13,16]), it accounts for an even greater proportion of the intracellular fluid (ICF) amino acid pool (~60%) and an extremely high proportion of the amino acid pool of muscle [2,15]. Figure 2 illustrates the central role of glutamine in nitrogen homeostasis and how the catabolic state can decrease intracellular (muscle) glutamine >50% and plasma glutamine levels 20–30%. Thus, in the catabolic state caused by cancer, nausea, cellular injury, or poor enteral glutamine intake, muscle glutamine stores may be called upon to provide glutamine to help to withstand and repair damage. This is because tissues are either glutamine consumers (high glutaminase activity in most organs and leucocytes and fibroblasts) or glutamine exporters (high glutamine synthetase activity in muscle and brain). 

The liver is a “switch-hitter” that is endowed with periportal hepatocytes that have high glutaminase activity to process glutamine from intestinal absorption and perivenous hepatocytes with high glutamine synthetase activity to provide glutamine to the plasma to nourish the GI-tract when enteral glutamine is lacking. This allows the GI tract to get a constant supply of glutamine to use as a respiratory fuel by balancing glutamine luminal absorption when enteral glutamine is plentiful with plasma extraction when it is not. Since skeletal muscle accounts for 38% of body mass in men and 30% in women [90], the contribution of muscle to glutamine homeostasis via the liver is a very significant one. Ammonia produced by intestinal glutaminase is cleared by the liver. Thus, both gut and liver contribute to glutamine homeostasis in the plasma and for mucosal surface metabolism (Figure 2).

Glutamine solubility is low (25 g/L); thus, suspensions are needed for topical oral and enteral supplementation; disaccharides can facilitate mucosal uptake [91]. Valencia et al. showed that in normal volunteers after ingestion of repeated daily high dose glutamine 0.3 g/kg/day, plasma glutamine remained at ~500 uM with only a minimal change (~20% increase) in plasma glutamine after this large daily dose after 10 days. This illustrates that the normally high concentration of glutamine is difficult to increase and probably of little relevance to tumors since glutamine is “always high” [92]. 

Glutamine supplementation without disaccharide has been also shown to have some benefit in other studies against OM associated with BMT [31,35,36]. Glutamine suspension with disaccharide to supply intraluminal glutamine for oral mucosa, esophagus, and intestines resulted serendipitously in not only a good-tasting formulation, but also a more effective treatment against mucositis than a glutamine-aspartame formulation. The glutamine-disaccharide combination facilitated >100-fold increased glutamine absorption by mucosal cells [68,91]. The glutamine-disaccharide formulation was effective against OM in a pilot study [26] as well as two randomized, placebo double-blind trials in cancer patients getting anthracycline based chemotherapy and autologous bone marrow transplant chemotherapy [27,28,30,68,93].

Trehalose, a disaccharide that not only facilitates glutamine cellular entry, but also has protein stabilizing properties and slower degradation to glucose than sucrose is contained in a commercially available glutamine-disaccharide product [93]. The cost of trehalose has become significantly less because it is now a commercially available sweetener (Treha, Cargill). Advantages of trehalose compared to sucrose for mucosal cell glutamine uptake include no oral degradation (less possibility of caries), stability in the acidic pH of the stomach, and once absorbed it is metabolized by the kidney so there is less of a glucose and insulin “bump” than the more rapid degradation and absorption for sucrose.

Enteric injury and diarrhea associated with abdominal radiation or chemotherapy can be ameliorated with glutamine supplementation. This should simultaneously increase glutathione in normal tissues such as the heart [84,94]. Glutamine supplementation may also decrease tumor glutaminase activity and tumor glutathione; this then may inhibit tumor utilization of glutamine for fuel and also make tumors more susceptible to damage from intracellular chemotherapy metabolites [80,81,83,85,87,95,96,97].

Irinotecan is a prodrug that is converted to the active metabolite SN-38 by intestinal bacteria. Its therapeutic effect is a blessing, but the severe, and sometimes delayed, prolonged enteritis from SN-38’s effects on the small bowel can adversely impact QOL. Delayed enteritis (e.g., days 5–15 after a 5–day cycle of oral irinotecan (50–90 mg/m^2^ daily for five days) is from chemotherapy-associated intestinal injury of the proximal intestine, which is exposed to the highest SN-38 concentrations. This delayed enteritis can be significantly reduced and sometimes completely eliminated by using the combination of glutamine with disaccharide during irinotecan and continuing for 7–10 days after oral irinotecan [93]. 

## 7. Is Glutamine Safe During Cancer Treatment 

Although normal and tumor bearing humans and animals have a high concentration of plasma glutamine (~500 uM) [13,16,92,98,99], does the tumor’s utilization of glutamine outweigh nutritional benefit? Glutamine is a supplement Generally Recognized as Safe (GRAS) by the FDA. The amount of glutamine in a normal diet is about 10 g/day but may be higher in a high protein diet [100]. Under catabolic conditions such as during and after chemotherapy and radiation therapy, it would be expected that there would be an increased need for glutamine in the diet and increased utilization by mucosal tissue as well as leucocytes. Mayers et al. recognized the importance of whole organism physiology to address the contribution of glutamine to not only disease, but also health maintenance in cancer [92]. For example, not enough protein and/or glutamine during cancer treatment would be expected to lead to sarcopenia and lymphopenia, more complications, and worse survival [7,8,101]. Adequate glutamine could be achieved with a high protein diet of 10–20 g/day, but 20–40 g may be needed if there is damage and stress [100]. If nausea is expected and/or there is impending or overt OM, esophagitis, or enteritis decreasing protein ingestion, ~4 g supplementary glutamine + disaccharide swish and swallow of the twice/day (BID) may be a safe and reasonable supplementary dose and schedule [93]. 

Although both glucose and glutamine are fuels for cancer cells by providing alpha ketoglutarate to the Krebs cycle to generate ATP [98,102], the contribution of tumor glutaminase to the Krebs cycle is probably much less than glucose. Glutamine can improve enteral nutrition, immune function and has differential effects on glutathione synthesis in tissue and tumors. One of the major challenges is knowing whether the glutamine level is always high in plasma and whether dietary modifications have any effect on glutamine metabolism at the site of tumors [11,102]. Overall, it would seem the benefits probably outweigh the deleterious effects of dietary protein including glutamine (summarized in Table 2). Furthermore, studies by Suzanne Klimberg’s group showed oral glutamine supplementation reduced toxicity from chemotherapy or radiation and had a protective effect in tumor bearing animals and was associated with improved survival Table 2 [11]. 

Furthermore, glutamine has also shown to decrease tumor incidence in mice genetically predisposed to develop squamous cell carcinoma [86]. The dynamic interplay between: mucosal /topical supplementation with a glutamine and disaccharide [93]; the enteral nutrition with protein adequate to avoid a catabolic state; the tendency to maintain a steady state plasma glutamine; and the intramuscular glutamine “reservoir” and liver glycogen stores is very complex and likely influenced by the intensity of cytotoxic therapy and radiation and the ongoing need for repair. Of all the amino acids that seem to ameliorate and/or prevent mucosal injury from chemotherapy or radiation, glutamine has been the most well studied. Because of its local effect, the glutamine swish and swallow approach is a simple but possibly significant step in ameliorating some of the toxic effects of cancer therapy and helping cancer patients keep nutritional goals on track. Table 3 details ways to supplement diets with glutamine.

## 8. A Balanced Approach to Nutrition During Cancer Therapy

Given the need to balance the metabolic needs of cancer patients for energy to avoid fatigue with lipids and amino acids to heal, food choices can be confusing. Basic underlying principles for caregivers should address timing, both in relation to chemotherapy and/or radiation, what is sustainable, and what are important macro and micronutrients. Although a common perception is that damage from chemotherapy and radiation is immediate, there is ongoing inflammation and damage and a need for more nutrition for 1–2 weeks after each chemotherapy cycle or end of radiation. It is often helpful to ask to see the RT treatment plan (like a contour map). This helps one know which normal tissues are at highest risk of mucosal damage (e.g., mouth, oropharynx, and esophagus). If a mucosal surface is “in field”, then review with a dietician and plan how to maintain enteral calorie and protein intake to pre-empt malnutrition problems and provide an opportunity for less toxicity. Keeping this in mind, the plans to facilitate enteral nutrition should continue for two to four weeks after completion of RT to promote additional healing. For chemotherapy side effects, a review of potential side effects and reassessment and making improvements after each cycle should be the goal. If mucositis, esophagitis, or enteritis is an issue, supplementation with glutamine and disaccharide is a convenient and relatively easy means to ameliorate mucosal damage.

## 9. Summary and Conclusions

Topical oral swish and swallow glutamine and a disaccharide, such as trehalose, has potential to ameliorate not only OM, but also esophagitis and enteritis after cancer chemotherapy and radiation. If cancer patients and caregivers recognize that it is possible to increase mucosal glutamine absorption using disaccharides, there may be less mucosal damage experienced by cancer patients. A small amino acid intervention may make a difference and possibly contribute to better overall nutritional status, improved survival with fewer complications, and ultimately less sarcopenia and lymphopenia.

## Figures and Tables

**Figure 1 nutrients-12-01675-f001:**
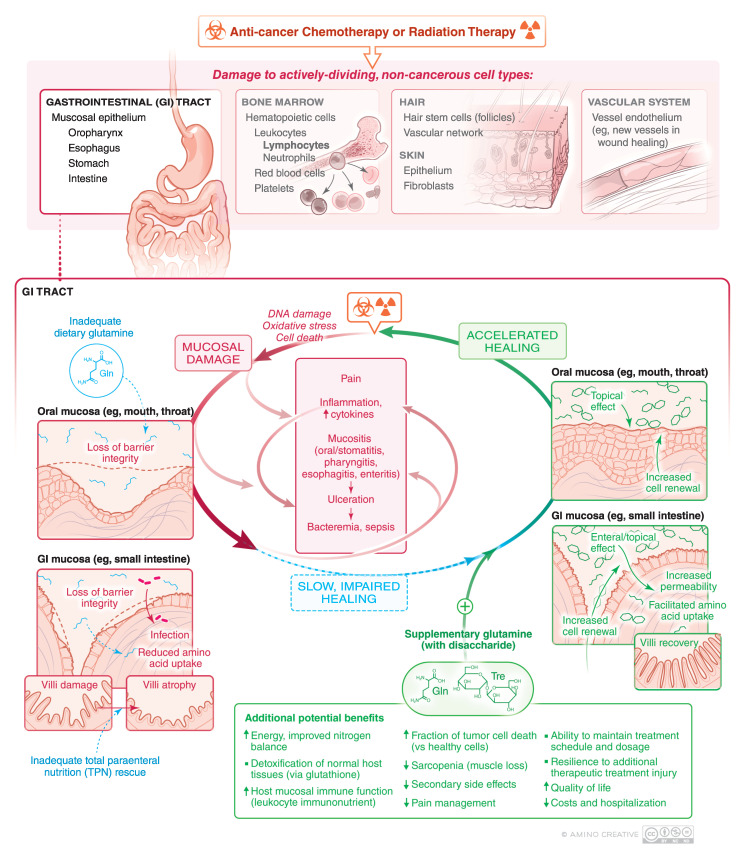
Physical damage and amino acid malnutrition can both contribute to slow healing and worse mucosal injury from cancer chemotherapy and/or radiation. Dietary glutamine may ameliorate some of these side effects of cancer therapy.

**Figure 2 nutrients-12-01675-f002:**
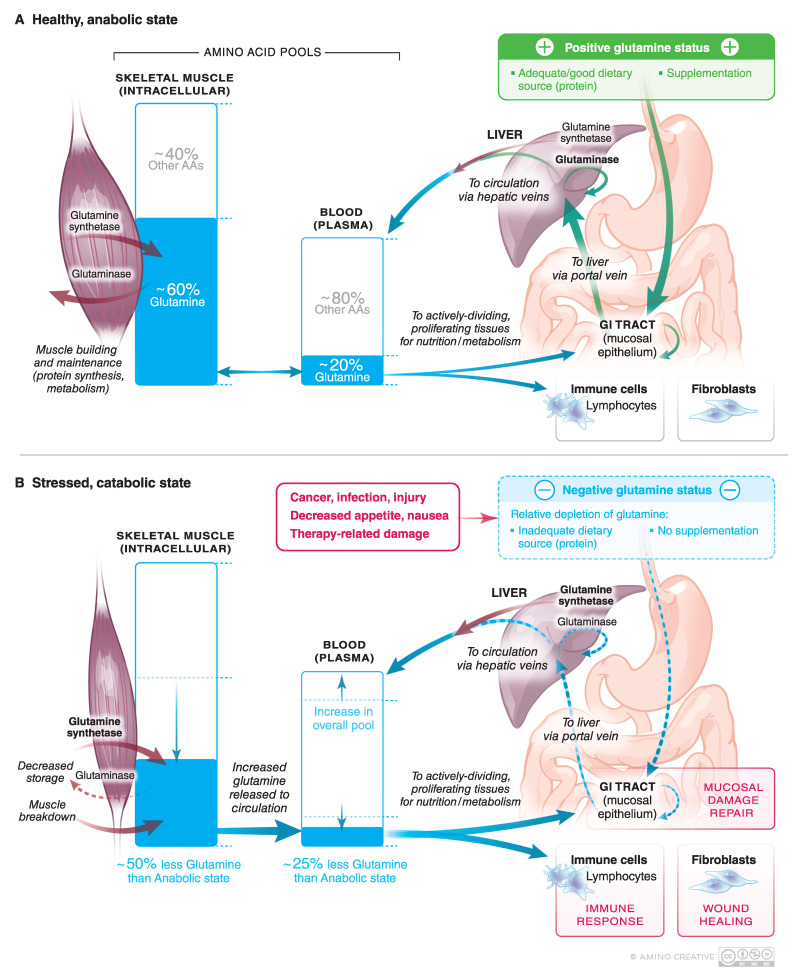
Pivotal role of glutamine stored in muscle (glutamine “bank”), normal high glutamine concentration in plasma, and liver amino acid metabolism (switch hitter) to facilitate steady state glutamine for mucosal health in (**A**) healthy anabolic state versus (**B**) during catabolic states including injury from cancer therapy, malnutrition, and tissue damage. In the catabolic state of mucositis from cancer therapy injury, topical glutamine + disaccharide can be helpful.

**Table 1 nutrients-12-01675-t001:** Chemotherapy drugs and radiation-associated^ side effects.

Chemotherapy Drug	Cytopenias *	Mucositis +/− GI **
Busulfan (BMT)	NLRP	2+
Carboplatin	NLRP	2+
Cisplatin	NLR	4+
Cyclophosphamide iv	NLRP	2+
Cyclophosphamide (oral)	NL	0
Cytarabine	NLRP	1+
Cytarabine (high dose)	NLRP	4+
Daunomycin	NLRP	3+
Dexamethasone	L	0
Docetaxel	NLRP	2+
Doxorubicin	NLRP	3+
Doxorubicin liposomes	minimal	2+
Etoposide (BMT)	NLRP	4+
Etoposide (oral)	NLRP	1+
Etoposide iv	NLRP	1+
Everolimus	L	2+
Gemcitabine	NLRP	1+
Ifosfamide	NLRP	0
Imatinib	minimal	0
Irinotecan	NLRP	4+
Melphalan (BMT)	NLRP	4+
Methotrexate (oral)	L	1+
Methotrexate (high dose)	NLRP	4+
NAB-Paclitaxel	NLRP	1+
Pazopanib	minimal	2+
Prednisone	L	0
^Radiation to GI tract	L	<where beam is>
(mucosal surfaces)		
Regorafenib	minimal	3+
Sirolimus (rapamycin)	L	1+
Sorafenib	NLRP	1+
Temozolomide	NLRP	0
Temsirolimus	NLRP	3+
Topotecan	NLRP	2+
Thiotepa (BMT)	NLRP	4+
^Total body Irradiation	NLRP	4+
Vincristine	L	0
Vinorelbine	NL	0

* Cytopenias: decreased N = Neutrophil, L = lymphocytes, R = red cells, P = platelets ** Mucositis + GI (anorexia, Nausea, vomiting, chemotherapy-associated diarrhea) 0 = none, 1 = occasional, 2 = common; 3 = very common 4 = severe.

**Table 2 nutrients-12-01675-t002:** Physiologic balancing act by glutamine. Damage Control + Tissue Regeneration and Tumor Suppression.

Information About Glutamine in Tumor Versus Treatment Effectiveness	Reference(s)
Major tumor fuel is glucose, but glutamine can also be used	[92,98,102]
Glutamine is a major fuel for lymphocytes and immune cells	[3,11]
Glutamine is needed by enterocytes to maintain intestinal health	[1]
Asparaginase which depletes glutamine in blood kills leukemia cells	[102,103,104]
Asparaginase w/o glutaminase activity is also highly effective	[105]
Glutamine effects glutathione levels (less in tumors, more in tissue)	[85,87,88,89]
Increased killing of tumors with glutamine supplementation	[85,95,106,107,108]
Glutamine is associated with less mucositis from chemotherapy	[26,27,28,65,68]
Glutamine reduces radiation side effects (helps normal tissues recover)	[42,66,80,81,82,93]
Glutamine’s decrease in squamous cell cancer incidence (cancer prevention)	[86]

**Table 3 nutrients-12-01675-t003:** Available supplements with glutamine in conjunction with cancer therapy.

Nutrients with High Glutamine	Supplement or Product (Source)
Dietary: foods high in glutamine	high protein foods (meats, fish, eggs, nuts, beans, milk)
High-protein milk + protein powder	FairLife milk + carnation breakfast (Nestle)
Water + protein powder	Beneprotein (Nestle), Whey powder, or plant protein powder
Protein Drinks	Boost, Ensure, Core Power, Pediasure, Peptamen
NG and G-tube formulas	Nutren (Nestle), Vital (Abbott), Jevity (Carewell)
Glutamine added to nutrient powder	Juven (Abbott)
Glutamine powder + maltodextrin	Glutasolve (Nestle)
Glutamine + sucrose+ trehalose	Healios (Enlivity)

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
