# Peer review of "Glutamine for Amelioration of Radiation and Chemotherapy Associated Mucositis during Cancer Therapy"

_nutrients, 2020, doi:10.3390/nu12061675_

Round 1

Reviewer 1 Report

The manuscript submitted for review concerns on the “Glutamine for Amelioration of Mucositis from Cancer Therapy” is very interesting. The whole review was extremely well documented by the authors. However, the authors need to be addressed the following concerns.

  1. The title of this review could be improved to make it clearer and more direct to the topic, since it’s lacking ‘Radiation Therapy’ concept.
  2. Abstract, line 1, delete the word “that”, its repeated twice.
  3. Abstract, line 6, the authors should have to mention the amount of glutamine as a diet supplement. Otherwise, need to revise the sentence make it clearer.
  4. Page 2, line 4, please replace “an amino acid” by “an α-amino acid”.
  5. Page 4, Table 1, Please check the caption, since authors mentioned both Chemotherapy and Radiation, but there is no details regarding ‘Radiation’ in tabular data.
  6. Page 10, line 10, changes “stability at acid” to “stable under acidic”.
  7. Page 10, paragraph 2, Please rewrite whole paragraph to make it more clear and logically.
  8. Page 10, paragraph 3, line 7, please correct “glutamine +disaccharide combination” instead of “combination of glutamine with disaccharide”.
  9. Page 12, paragraph 2, line 11, Changes “With this knowledge and plan in place” to “Keeping with this in mind, and”.
  10. Please double check all the references to ensure, they are correctly and followed as per the article guidelines.

Author Response

Author response to Reviewer 1. Thank-you for the suggestions. These have improved the manuscript and are as follows:

  1. The title of this review could be improved to make it clearer and more direct to the topic, since it’s lacking ‘Radiation Therapy’ concept.

Suggested title: “ Glutamine for Amelioration of Radiation and Chemotherapy Associated Mucositis During Cancer Therapy”

  1. Abstract, line 1, delete the word “that”, its repeated twice

Suggestions=done

  1. Abstract, line 6, the authors should have to mention the amount of glutamine as a diet supplement. Otherwise, need to revise the sentence make it clearer.

Added 10 gm/day as amount

  1. Page 2, line 4, please replace “an amino acid” by “an α-amino acid”.

Added L-alpha-amino acid

  1. Page 4, Table 1, Please check the caption, since authors mentioned both Chemotherapy and Radiation, but there is no details regarding ‘Radiation’ in tabular data

Details in table 1 have been expanded to radiation to GI tract and TBI expanded to total body irradiation * with formatting to make radiation more easily appreciate has been done so title is more accurate, as requested

  1. Page 10, line 10, changes “stability at acid” to “stable under acidic”.

Suggestion = done

  1. Page 10, paragraph 2, Please rewrite whole paragraph to make it more clear and logically.

Thank-you!  The paragraph has been revised to be more clear and logical as follows:

“Enteric injury and diarrhea associated with abdominal radiation or chemotherapy can be ameliorated with glutamine supplementation.  This should simultaneously increase glutathione in normal tissues such as the heart [84, 94]. Glutamine supplementation may also decrease tumor glutaminase activity and tumor glutathione; this then may inhibit tumor utilization of glutamine for fuel and also make tumors more susceptible to damage from intracellular chemotherapy metabolites)  [78, 80, 81, 83, 85, 87, 95-97]. “

  1. Page 10, paragraph 3, line 7, please correct “glutamine +disaccharide combination” instead of “combination of glutamine with disaccharide”.

Suggestions done.

  1. Page 12, paragraph 2, line 11, Changes “With this knowledge and plan in place” to “Keeping with this in mind, and”.

Suggestion= done

  1. Please double check all the references to ensure, they are correctly and followed as per the article guidelines.

References re-checked and formatted using EndNote

Reviewer 2 Report

The authors compiled a describing the implications of Glutamine for amelioration of Mucositis from cancer therapy. It is a very well-put together review. I enjoyed reading it. My only suggestion would be to include a tabular form of clinically approved Gultamine supplementations that might be used in conjuction with cancer therapy. 

Author Response

Author reply to reviewer 2

My only suggestion would be to include a tabular form of clinically approved glutamine supplementations that might be used in conjunction with cancer therapy. 

Author response:

Since glutamine is a natural L-alpha amino acid nutrient in food (and not a clinically approved drug). we have added text and a table 3 as suggested by reviewer 2.

We very much appreciate the recommendation and are open to additional suggestions and formatting of the table as this may become an important educational aid for nutritionists and caregivers seeking to supplement the diet of cancer patients with L-glutamine.

      Table 3.  Available Supplements with Glutamine in Conjunction with Cancer Therapy

Nutrients with high glutamine                  Supplement or Product (source)

Dietary: foods high in glutamine              high protein foods (meats, fish, eggs, nuts, beans, milk)

High-protein milk+ protein powder          FairLife milk + carnation breakfast (Nestle)

Water + protein powder                         Beneprotein (Nestle), Whey powder, or plant protein powder

Protein Drinks                                       Boost, Ensure, Core Power, Pediasure, Peptamen

NG and G-tube formulas                        Nutren (Nestle), Vital (Abbott), Jevity (Carewell)

Glutamine added to nutrient powder       Juven (Abbott)

Glutamine powder+ maltodextrin            Glutasolve (Nestle)

Glutamine + sucrose+ trehalose             Healios (Enlivity)
